# PLUG-AND-PLAY PROMPT REFINEMENT VIA LATENT FEEDBACK FOR DIFFUSION MODEL ALIGNMENT

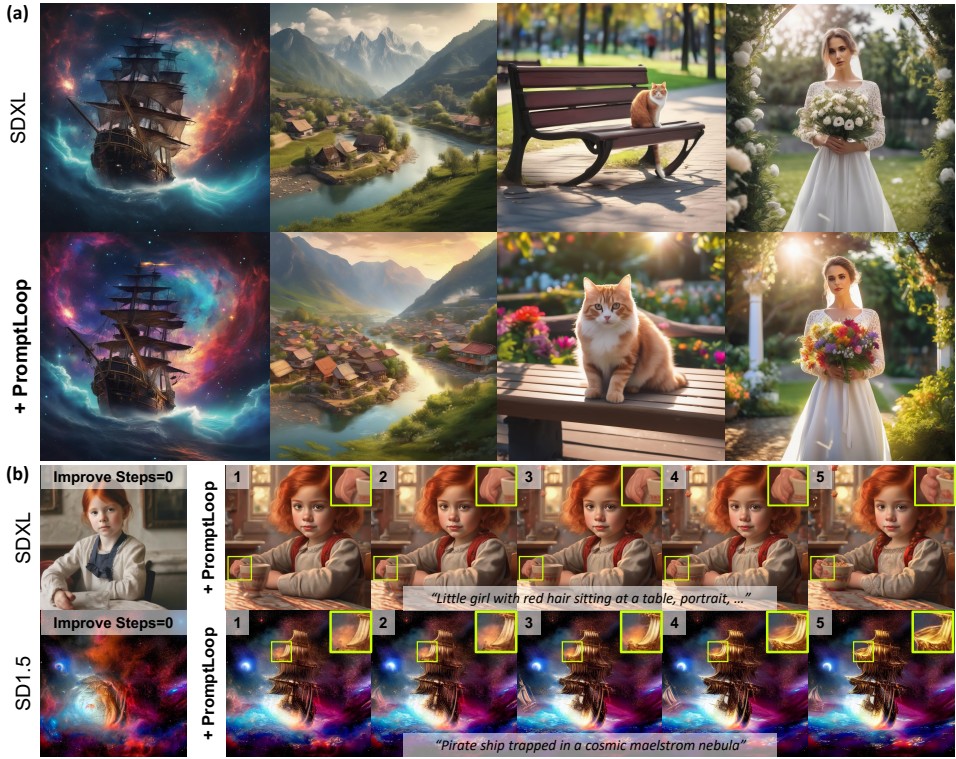

Figure 1: (a) PromptLoop uses latent feedback for stepwise prompt refinement, achieving functional equivalence to diffusion model RL and effective reward alignment (shown with ImageReward). (b) Multiple timestep-aware prompt updates during a single sampling yield stronger alignment.

## ABSTRACT

Despite the recent progress, reinforcement learning (RL)-based fine-tuning of diffusion models often struggles with generalization, composability, and robustness against reward hacking. Recent studies have explored prompt refinement as a modular alternative, but most adopt a feed-forward approach that applies a single refined prompt throughout the entire sampling trajectory, thereby failing to fully leverage the sequential nature of reinforcement learning. To address this, here we introduce *PromptLoop*, a plug-and-play RL framework that incorporates latent feedback into step-wise prompt refinement. Rather than modifying diffusion model weights, a multimodal large language model (MLLM) is trained with RL to iteratively update prompts based on intermediate latent states of diffusion models. This design achieves a structural analogy to the Diffusion RL approach, while retaining the flexibility and generality of prompt-based alignment. Extensive experiments across diverse reward functions and diffusion backbones demonstrate that PromptLoop (i) achieves effective reward optimization, (ii) generalizes seamlessly to unseen models, (iii) composes orthogonally with existing alignment methods, and (iv) mitigates over-optimization and reward hacking.

# 1 INTRODUCTION

Diffusion models (Ho et al., 2020; Song et al., 2020b; Rombach et al., 2022) have now become the state of the art for image generation. Recently, increasing attention has been directed toward reinforcement learning (RL)-based approaches (Sutton et al., 1998) that align these models with user preferences through explicit reward optimization. Algorithms such as PPO (Schulman et al., 2017) and DPO (Rafailov et al., 2023) have been applied directly to fine-tune diffusion model parameters (Black et al., 2024; Wallace et al., 2024). With reward functions defined over aesthetic quality, safety, human preference, or prompt alignment, these methods successfully steer model behavior without requiring new training data. However, direct RL fine-tuning remains limited: improvements often fail to generalize across models, additional enhancements are not easily composable once fine-tuning is complete, and pathological behaviors such as reward hacking or over-optimization can arise (Kim et al., 2025b).

In parallel, the rapid development of large language models (LLMs) (Brown et al., 2020; Grattafiori et al., 2024; Guo et al., 2025) and multimodal large language models (MLLMs)(Liu et al., 2023; Wang et al., 2024a; 2025b) has inspired a new research direction: refining the input prompts rather than the diffusion model itself. These prompt-alignment methods either guide an LLM to improve a user's prompt or adopt iterative feedback loops for prompt refinement (Mañas et al., 2024; Kim et al., 2025a; Khan et al., 2025). Building further, Hao et al. (2023) and Wu et al. (2025) propose to fine-tune LLMs with RL, enabling them to generate goal-directed prompt modifications more effectively. Compared to weight-level tuning, prompt refinement is attractive because prompts are shared across all text-to-image (T2I) models, inherently supporting generalization and orthogonal composability. Moreover, prompts, being abstract and discrete, may act as a buffer against reward hacking by decoupling reward optimization from direct parameter updates (Lester et al., 2021; Xie et al., 2022; Genewein et al., 2025). For a detailed discussion of related works, see Appendix A. Nevertheless, prompt-based strategies remain structurally distinct from weight-level approaches. In diffusion models, parameters interact directly with intermediate latent variables $x_t$ in a feedback loop, where each denoising step conditions on $x_t$ to produce $x_{t-1}$. By contrast, existing RL-based prompt refinement methods typically operate in a feed-forward manner, producing a refined prompt once and applying it uniformly across all timesteps, without leveraging the evolving latent trajectory.

To bridge this gap, we propose a generalized RL-based reward alignment framework called *Prompt-Loop* that achieves structural analogy to weight-level fine-tuning while preserving the modularity of prompt refinement (Fig. 2). Specifically, our method introduces a plug-and-play prompt refinement module as a policy. This module leverages a MLLM to process feedback from the intermediate latent $x_t$ as one of the states, analogous to diffusion RL formulations, and then refines the prompt $c_t$ as the action injected into subsequent denoising steps. Thus, the sampling dynamics are adaptively adjusted without direct fine-tuning of the diffusion model itself. Unlike approaches that either delay feedback until after sampling or confine it to external loops, our method adopts a diffusion RL–style closed-loop design that embeds refinement directly within a single diffusion pass, ultimately enabling fine-grained adaptive control and improved efficiency. Extensive experiments across diverse diffusion models and reward functions demonstrate that our approach not only achieves effective reward optimization, but also generalizes robustly to unseen models, composes orthogonally with existing alignment methods, and mitigates over-optimization and reward hacking. These results establish PromptLoop as a practical and versatile approach to reward alignment for diffusion models.

Our contributions are summarized as follows:

- PrompLoops incorporates step-wise latent feedback into prompt refinement, achieving structural analogy to parameter-level tuning without modifying model weights.

- We demonstrate broad generalization, effective reward optimization, and mitigation of reward hacking across diverse models and reward functions.

# 2 PRELIMINARIES

**Diffusion Models.** Diffusion models (Ho et al., 2020; Song & Ermon, 2019; Sohl-Dickstein et al., 2015) are a class of latent variable generative models that approximate the data distribution $x_0 \sim$

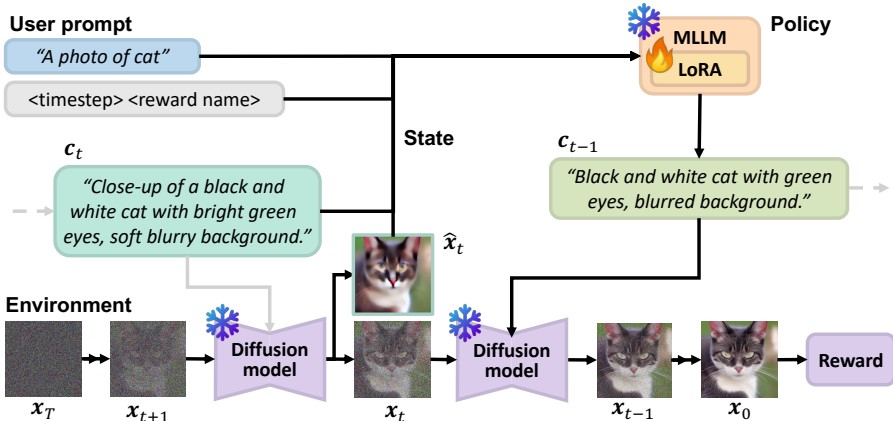

Figure 2: **Closed-loop prompt refinement framework with RL.** At each denoising step, the policy MLLM takes the current state—posterior estimates, the user query, and prior refinements—and generates an action, a refined prompt. The diffusion model then updates the state, and this loop continues until the final image is produced and scored by the reward model.

$p_{\text{data}}$ through a hierarchical latent process. The generative distribution is formulated as

$$p_\phi(\boldsymbol{x}_0) = \int p(\boldsymbol{x}_T) \prod_{t=1}^{T} p_\phi^{(t)}(\boldsymbol{x}_{t-1}|\boldsymbol{x}_t) \, d\boldsymbol{x}_{1:T}, \tag{1}$$

where the prior $p(\boldsymbol{x}_T)$ is typically a standard Gaussian distribution. The latent sequence $\{\boldsymbol{x}_t\}_{t=1}^T$ is obtained via a forward noising process, which follows a Markov chain with a variance schedule $\{\beta_t\}_{t=1}^T$:

$$q(\boldsymbol{x}_t|\boldsymbol{x}_{t-1}) = \mathcal{N}(\boldsymbol{x}_t \mid \sqrt{\alpha_t}\boldsymbol{x}_{t-1}, (1-\alpha_t)I), \quad q(\boldsymbol{x}_t|\boldsymbol{x}_0) = \mathcal{N}(\boldsymbol{x}_t \mid \sqrt{\bar{\alpha}_t}\boldsymbol{x}_0, (1-\bar{\alpha}_t)I), \tag{2}$$

where $\alpha_t = 1 - \beta_t$ and $\bar{\alpha}_t = \prod_{i=1}^{t} \alpha_i$. Training is carried out by learning to predict the injected Gaussian noise $\boldsymbol{\epsilon}$ using a neural network $\hat{\boldsymbol{\epsilon}}_\phi$, which is often conditioned by $\boldsymbol{c}$, known as $\epsilon$-*matching*. This is equivalent to denoising score matching (DSM) (Vincent, 2011; Song & Ermon, 2019), which estimates the score function $\nabla_{\boldsymbol{x}_t} \log p(\boldsymbol{x}_t)$:

$$\mathcal{L}_{\epsilon-\text{matching}} = \mathbb{E}_{t,\boldsymbol{x}_0,\boldsymbol{\epsilon}}\left[ \|\hat{\boldsymbol{\epsilon}}_\phi(\boldsymbol{x}_t, t, \boldsymbol{c}) - \boldsymbol{\epsilon}\|_2^2 \right], \tag{3}$$

where $\boldsymbol{x}_t = \sqrt{\bar{\alpha}_t}\boldsymbol{x}_0 + \sqrt{1-\bar{\alpha}_t}\,\boldsymbol{\epsilon}$ with $\boldsymbol{\epsilon} \sim \mathcal{N}(0, I)$. Once trained, the model iteratively reverses the noising process as follows:

$$\boldsymbol{x}_{t-1} = f(\boldsymbol{x}_t, \boldsymbol{z}_t, \boldsymbol{c}, t) := \frac{1}{\sqrt{\alpha_t}}\left( \boldsymbol{x}_t - \frac{1-\alpha_t}{\sqrt{1-\bar{\alpha}_t}}\hat{\boldsymbol{\epsilon}}_\phi(\boldsymbol{x}_t, t, \boldsymbol{c}) \right) + \sigma_t \boldsymbol{z}_t, \tag{4}$$

where $\boldsymbol{z}_t \sim \mathcal{N}(0, I)$ and $\sigma_t^2 = \frac{1-\bar{\alpha}_{t-1}}{1-\bar{\alpha}_t}\beta_t$. This corresponds to the canonical DDPM sampler (Ho et al., 2020). In general, $f(\cdot)$ can be replaced by a variety of alternative samplers such as DDIM (Song et al., 2020a), PNDM (Liu et al., 2022), Euler (Karras et al., 2022), DPM-solver (Lu et al., 2022).

## 3 PROMPTLOOP

### 3.1 MDP FORMULATION

In PromptLoop, as shown in Fig. 2, we aim to generate a refined text prompt $\boldsymbol{c}_{t-1}$ conditioned on user input $q$ and interpret intermediate visual states $\boldsymbol{x}_t$ arising during the reverse diffusion process. The refined text prompt is then used to generate the next visual sample $\boldsymbol{x}_{t-1}$. To this end, we adopt a multimodal language model (MLLM) (Liu et al., 2023; Wang et al., 2024a; 2025b) that accepts multimodal inputs and outputs refined prompts at each timestep. Then, our RL framework is to

| | Diffusion RL | PromptLoop (Ours) |
|---|---|---|
| State $s_t$ | $(\boldsymbol{x}_t, q, t)$ | $(\boldsymbol{x}_t, \boldsymbol{c}_t, q, t)$ |
| Policy | Diffusion model $p_\phi$ | VLLM $\pi_\theta$ |
| Action $a_t$ | $\boldsymbol{x}_{t-1} \sim p_\phi(\cdot \vert s_t)$ | $\boldsymbol{c}_{t-1} \sim \pi_\theta(\cdot \vert s_t)$ |
| Transition | – | $\boldsymbol{x}_{t-1} = f(\boldsymbol{x}_t, \boldsymbol{z}_t, \boldsymbol{c}_{t-1}, t)$ |
| Reward $R$ | $r(\boldsymbol{x}_0, q)$ | $r(\boldsymbol{x}_0, q)$ |

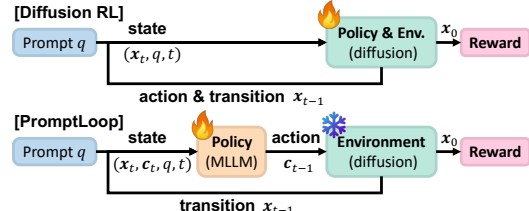

Table 1: Structural analogy and key differences in MDP formulation between Diffusion RL and our proposed PromptLoop framework.

Figure 3: Latent feedback establishes a functional correspondence with Diffusion RL, while Prompt-Loop diverges by adjusting the diffusion dynamics through time-step aware prompts as actions.

train the MLLM to maximize the reward at the final visual state $\boldsymbol{x}_0$. Formally, our Markov decision process (MDP) is defined as the $T$-step reverse process with state $s_t$ and actions $a_t$:

$$s_t = (\boldsymbol{x}_t, \boldsymbol{c}_t, q, t), \quad a_t = \boldsymbol{c}_{t-1}, \tag{5}$$

which are conditioned on an initial user prompt $q$, a previously updated prompt $\boldsymbol{c}_t$, and a visual state $\boldsymbol{x}_t$. Then, an action is sampled from the *MLLM policy* as $a_t \sim \pi_\theta(\cdot \mid s_t)$ and the visual state $\boldsymbol{x}_{t-1}$ is updated using the frozen diffusion model with the updated prompt $\boldsymbol{c}_{t-1}$. A terminal reward $r(\boldsymbol{x}_0, q)$ is assigned at the final step.

This is in contrast to directly training the diffusion model's parameters (Black et al., 2024; Wallace et al., 2024) using RL, where MDP is defined with the state and action:

$$s_t = (\boldsymbol{x}_t, q, t), \quad a_t = \boldsymbol{x}_{t-1} \tag{6}$$

where an action is sampled from the *diffusion* policy $\boldsymbol{x}_t \sim p_\phi(\cdot \vert s_t)$. The difference between the original Diffusion-RL and our RL framework is detailed in Tab. 1 and Fig. 3.

Note that our MDP formulation provides a structural correspondence between diffusion-model-based RL and the prompt refinement framework, enabled by a time-step-aware closed-loop latent feedback mechanism. On the other hand, in direct fine-tuning of diffusion models using RL, the diffusion model should be trained as the optimization target. This direct RL fine-tuning remains limited: improvements often fail to generalize across models, additional enhancements are not easily composable once fine-tuning is complete, and pathological behaviors such as reward hacking or over-optimization can arise. In our framework, the timestep-aware prompt-level actions can approximate the functional role of weight-level control, while retaining plug-and-play modularity, generalization, composability, and robustness against reward hacking.

Furthermore, our approach has fundamental advantages over other prompt finetuning approaches. Specifically, prior prompt-tuning approaches either lack an intrinsic feedback loop (Hao et al., 2023; Wu et al., 2025; Wang et al., 2025a) or deliver feedback only after a full sampling (Mañas et al., 2024; Kim et al., 2025a; Khan et al., 2025), making them fundamentally different from our MDP formulation.

## 3.2 OPTIMIZATION

At the end of each episode (*i.e.*, $\boldsymbol{x}_T, \boldsymbol{x}_{T-1}, \ldots, \boldsymbol{x}_0$), the fully generated image $\boldsymbol{x}_0$ is evaluated by a reward function $r$ to produce a reward $R = r(\boldsymbol{x}_0, q)$. This can encode diverse criteria such as aesthetic quality (Schuhmann, 2025), safety (LAION-AI, 2023), prompt alignment (Radford et al., 2021), or human preference (Wu et al., 2023; Xu et al., 2023). The diffusion model and the reward model are both treated as black-box components: no gradient flows through them, and the policy is updated solely based on observed rewards.

Policy gradient methods (Williams, 1992; Sutton et al., 1999) optimize this objective by estimating gradients with respect to $\theta$. A widely used algorithm is Proximal Policy Optimization (PPO) (Schulman et al., 2017), which improves stability by constraining policy updates through a clipped surro-

gate objective:

$$\mathcal{L}_{\text{PPO}}(\theta) = \mathbb{E}_t \left[ \min\left( \rho_t(\theta) \, \hat{A}_t, \ \text{clip}(\rho_t(\theta), 1 - \epsilon, 1 + \epsilon) \, \hat{A}_t \right) \right] - \beta \, \text{KL}[\pi_{\theta_{\text{old}}}(\cdot \mid s_t) \, \| \, \pi_\theta(\cdot \mid s_t)],$$

$$\text{where} \quad \rho_t(\theta) = \frac{\pi_\theta(a_t \mid s_t)}{\pi_{\theta_{\text{old}}}(a_t \mid s_t)}.$$

$$(7)$$

Here, $\beta$ is a hyperparameter controlling the strength of the KL penalty, and the advantage $\hat{A}_t$ measures how much better an action is than the expected value under the current policy. Especially, Group Relative Policy Optimization (GRPO) (Guo et al., 2025) replaces the advantage estimator with a group-normalized reward to stabilize training and reduce variance:

$$A_i = \frac{r_i - \text{mean}(\{r_j(\cdot)\}_{j=1}^G)}{\text{std}(\{r_j(\cdot)\}_{j=1}^G)},$$

$$(8)$$

where $\{r_j(\cdot)\}_{j=1}^G$ are the rewards of $G$ sampled outputs for the same prompt. Therefore, we employ the standard token-level Group Relative Policy Optimization (GRPO) (Guo et al., 2025). Each training episode is initialized with user prompts drawn from a prompt-only dataset and proceeds via an online, on-policy reinforcement learning procedure.

### 3.3 IMPLEMENTATION

As part of our implementation, we design the MLLM's input to be denoised latent representations rather than raw noisy states $x_t$. Specifically, we convert the noisy visual latent state $x_t$ into its denoised estimate $\hat{x}_t$, which lies closer to the data manifold and thus provides a more semantically meaningful input to the policy model (Chung et al., 2022; Yu et al., 2023):

$$\hat{x}_t = \frac{1}{\sqrt{\bar{\alpha}_t}} \big( x_{t+1} - \sqrt{1 - \bar{\alpha}_t} \, \hat{\epsilon}_\phi(x_{t+1}, c_t, t) \big).$$

$$(9)$$

While our framework achieves structural equivalence, it introduces an additional computational overhead: the policy model must be invoked during every denoising step of the diffusion process. This requirement also significantly increases memory costs, as both the diffusion model and the policy MLLM must be co-resident on the accelerator (*e.g.* VRAM), or alternatively, incur large transfer times under offloading. Such constraints not only limit practical applicability but also complicate the seamless integration of our approach into existing user-level diffusion-based image generation pipelines.

To mitigate these issues, we adopt a sparse refinement strategy, where *prompt refinement steps* are defined as a set of timesteps $\mathcal{R} \subseteq \{1, \ldots, T\}$ with $|\mathcal{R}| = N_R$. The policy model is applied only at these steps rather than at every denoising step. For example, if the policy refines the prompt at timestep $t_1$ and the next refinement occurs at $t_2$ with $t_1 > t_2$, then $c_{t_1-1:t_2} = \pi_\theta(\cdot \mid s_{t_1})$ and remains fixed until the next refinement step. During training, $\mathcal{R}$ is sampled uniformly at random, while during inference it is deterministically set at even intervals. This design allows the policy to generalize to an arbitrary number of refinement steps during sampling.

We empirically observe that visual feedback from intermediate denoised states—though essential during training—is not strictly necessary at inference. Once the policy has learned the transition dynamics of the environment (*i.e.*, the diffusion process coupled with the reward model), it can generate effective refinements without explicit access to intermediate visual signals. Consequently, refined prompts for all timesteps can be generated *a priori*, allowing the diffusion process to proceed without interruptions during inference. This design yields substantial generalization capability and efficiency gains while remaining fully compatible with existing diffusion model ecosystems, requiring no modification to the generation loop and offering the same ease of integration as feed-forward prompt optimization methods, yet uniquely retaining the advantages of closed-loop RL fine-tuning.

## 4 EXPERIMENTAL RESULTS

### 4.1 METHODS

**Tasks.** To evaluate our framework as a general black-box reward alignment system, we consider two categories of reward models: *single reward* and *composite reward*. For the single reward setting, we

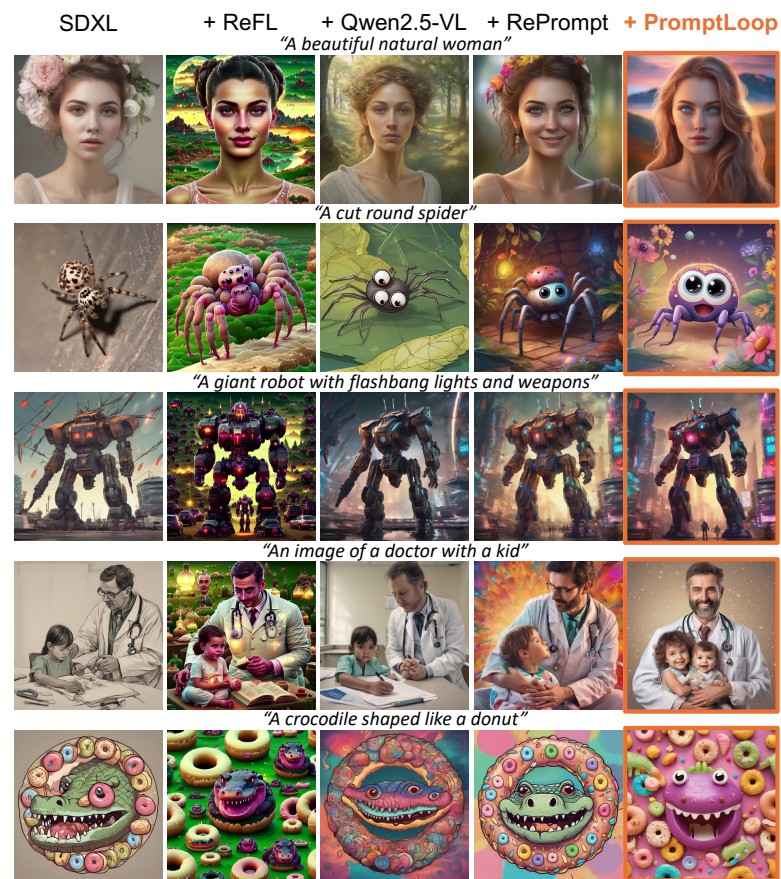

Figure 4: Qualitative comparison of single-reward alignment, illustrating improvements over baseline methods. (SDXL & ImageReward)

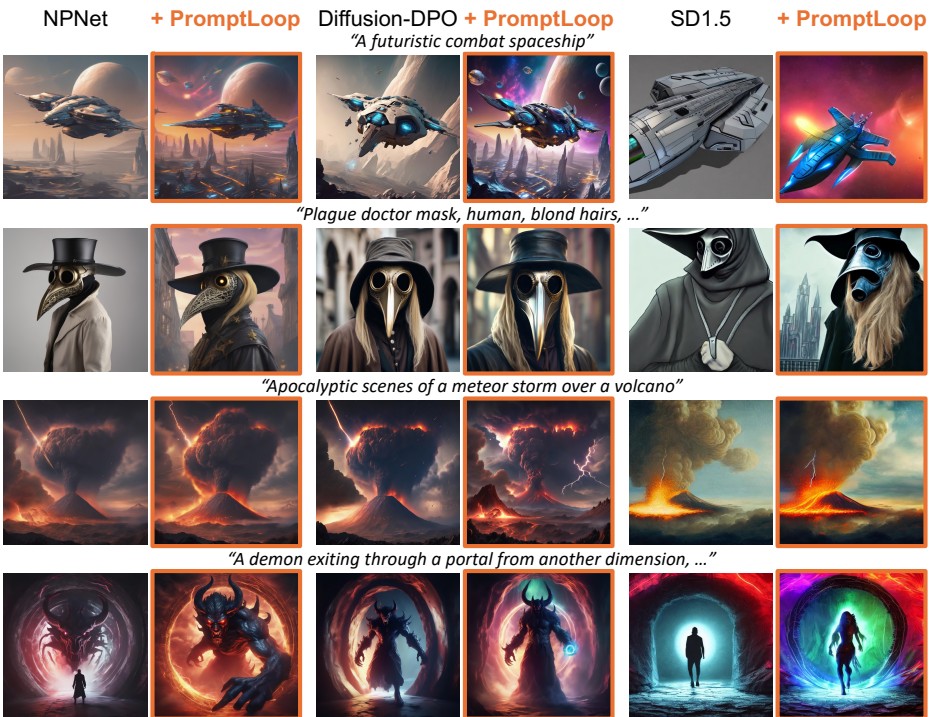

Figure 5: Qualitative results showing the orthogonality and generalizability achieved by applying our method to unseen reward-alignment baselines (SDXL & ImageReward).

Table 2: Quantitative evaluation on single-reward alignment with SD1.5 and SDXL, showing comparison with baselines and demonstrating orthogonality and generalizability.

| Training setup | Method | ImageReward | HPSv2 | Aesthetics | VLLM Score |
|---|---|---|---|---|---|
| SDXL & ImageReward | SDXL | 0.7244 | 0.2805 | 6.073 | 0.735 |
| | + ReFL (Xu et al., 2023) | 1.0119 | 0.2740 | 6.286 | 0.715 |
| | + Qwen2.5-VL-3B (Bai et al., 2025) | 0.5114 | 0.2739 | 6.279 | 0.741 |
| | + RePrompt | 1.0148 | 0.2796 | 6.518 | 0.763 |
| | **+ PromptLoop (ours)** | **1.0948** | **0.2807** | **6.583** | **0.764** |
| | SDXL + Diffusion-DPO (Wallace et al., 2024) | 0.9921 | **0.2868** | 6.015 | 0.731 |
| | **+ PromptLoop (ours)** | **1.2898** | 0.2862 | **6.491** | **0.763** |
| | SDXL + NPNet (Zhou et al., 2025) | 0.7357 | 0.2805 | 6.059 | 0.733 |
| | **+ PromptLoop (ours)** | **1.1213** | **0.2811** | **6.561** | **0.762** |
| | SD1.5 (Rombach et al., 2022) | 0.0816 | 0.2678 | 5.458 | 0.675 |
| | **+ PromptLoop (ours)** | **0.4546** | **0.2688** | **5.813** | **0.723** |
| SD1.5 & ImageReward | SD1.5 | 0.0816 | 0.2678 | 5.458 | 0.675 |
| | + DDPO (Black et al., 2024) | 0.6051 | **0.2726** | 5.562 | 0.693 |
| | + ReFL (Xu et al., 2023) | 0.6248 | **0.2748** | 5.577 | 0.691 |
| | + Qwen2.5-VL-3B (Bai et al., 2025) | -0.1720 | 0.2628 | 5.668 | 0.693 |
| | + RePrompt | 0.4344 | 0.2684 | 5.850 | 0.722 |
| | **+ PromptLoop (ours)** | **0.6320** | 0.2701 | **5.853** | **0.725** |
| | SD1.5 + DDPO (Black et al., 2024) | 0.6051 | 0.2726 | 5.562 | 0.693 |
| | **+ PromptLoop (ours)** | **0.9842** | **0.2742** | **5.926** | **0.726** |
| | SD1.5 + Diffusion-DPO (Wallace et al., 2024) | 0.3012 | 0.2717 | 5.568 | 0.687 |
| | **+ PromptLoop (ours)** | **0.7920** | **0.2739** | **5.968** | **0.734** |
| | SD1.5 + ReFL (Xu et al., 2023) | 0.6248 | 0.2748 | 5.577 | 0.691 |
| | **+ PromptLoop (ours)** | **0.9271** | **0.2751** | **5.877** | **0.724** |
| | SDXL (Podell et al., 2023) | 0.7244 | 0.2805 | 6.073 | 0.735 |
| | **+ PromptLoop (ours)** | **1.0859** | **0.2807** | **6.535** | **0.763** |

Table 3: Quantitative evaluation on composite-reward alignment with SDXL-turbo, showing comparison with baselines and demonstrating orthogonality and generalizability.

| Training setup | Method | GenEval | ImageReward | HPSv2 |
|---|---|---|---|---|
| SDXL-turbo & RePrompt | SDXL-turbo (Sauer et al., 2024) | 0.5445 | 0.7769 | 0.2915 |
| | + Qwen2.5-VL-3B (Bai et al., 2025) | 0.5212 | 0.6417 | 0.2893 |
| | + RePrompt (Wu et al., 2025) | 0.5101 | 0.7876 | 0.2912 |
| | **+ PromptLoop (ours)** | **0.5483** | **0.8516** | **0.2938** |
| | SDXL (Podell et al., 2023) | 0.5431 | 0.5518 | 0.2886 |
| | **+ PromptLoop (ours)** | **0.5505** | **0.7420** | **0.2906** |
| | SD1.5 (Rombach et al., 2022) | 0.4206 | -0.1315 | 0.2783 |
| | **+ PromptLoop (ours)** | **0.4399** | **-0.0375** | **0.2793** |

adopt ImageReward (Xu et al., 2023), a widely used neural network–based reward function for human preference and prompt alignment, along with incompressibility, compressibility, and aesthetic score models (Black et al., 2024; Schuhmann, 2025). These rewards are applied to train Stable Diffusion v1.5 (Rombach et al., 2022) (SD1.5) and Stable Diffusion XL (Podell et al., 2023) (SDXL) using prompts from the Pick-a-Pic v2 dataset (Kirstain et al., 2023). For the composite reward setting, we follow a RePrompt-style design (Wu et al., 2025), which combines ImageReward, VLLM-reward (OpenAI, 2025), and additional task-specific signals such as format and length reward. This composite reward style is intended to better capture human preference and object-focused alignment. Compared to the single reward setting, the composite reward is more complex and difficult to optimize, since it requires balancing multiple heterogeneous objectives simultaneously. We use it to train Stable Diffusion XL Turbo (Sauer et al., 2024) (SDXL-turbo),a distillation model designed for few-step generation, with the prompt dataset introduced by Wu et al. (2025).

**Evaluations.** In evaluation, we validate our model's capability along three aspects: performance, orthogonality, and generalizability. For performance evaluation, we compare against baseline reward alignment methods, including DDPO (Black et al., 2024), ReFL (Xu et al., 2023), Qwen2.5-VL-3B (Bai et al., 2025), and RePrompt (Wu et al., 2025). For orthogonality, we apply our trained

Table 4: Ablation study results showing the effectiveness of each proposed component.

| Components | ImageReward | HPSv2 | VLLM Score |
|---|---|---|---|
| SD1.5 | 0.0816 | 0.2678 | 0.675 |
| + policy model | -0.2315 | 0.2617 | 0.681 |
| + GRPO training | 0.4344 | 0.2684 | 0.722 |
| + multiple improvement | 0.4912 | 0.2690 | 0.724 |
| + visual feedback | **0.6320** | **0.2701** | **0.725** |

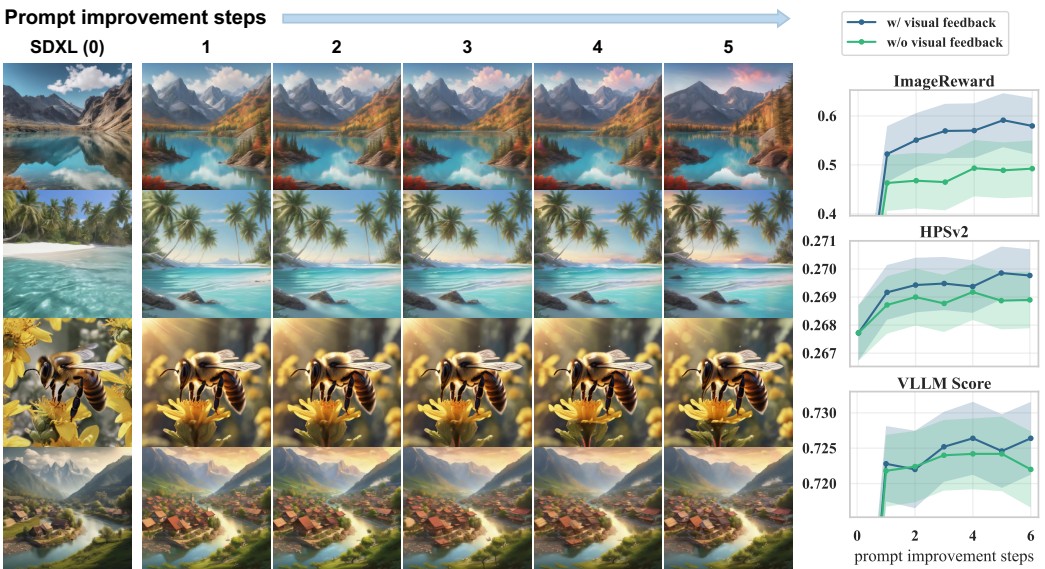

Figure 6: Ablation study demonstrating that incorporating visual feedback and increasing the number of refinement steps consistently enhances reward alignment. (Left: SDXL, Right: SD1.5; reward: ImageReward)

policy model to other diffusion models that are fine-tuned or augmented with additional modules for human preference alignment, demonstrating that our method can be applied orthogonally to existing preference alignment techniques. Specifically, we evaluate on DDPO (Black et al., 2024), Diffusion-DPO (Wallace et al., 2024), ReFL, and NPNet (Zhou et al., 2025). These experiments demonstrate that our method can be applied orthogonally to diverse alignment techniques without requiring retraining. For generalizability, we evaluate our trained policy model on different versions of text-to-image diffusion models that were not seen during training. It is important to note that for both orthogonality and generalizability, the policy model was only trained on the vanilla diffusion model environment, which differs from the sampling variants.

## 4.2 RESULTS

**Single Reward.** After aligning SD1.5 and SDXL models with the ImageReward reward function, we conducted quantitative evaluations (Tab. 2). The results demonstrate that our proposed methodology consistently outperforms baselines not only with respect to the target reward, but also across most evaluation metrics. Crucially, our method is orthogonal, demonstrating broad compatibility with a variety of human preference alignment strategies—including noise optimization, reinforcement learning, and gradient-based optimization—regardless of their internal mechanisms. While it may underperform on specific standalone metrics, its strength lies in complementing and enhancing existing baselines.

The qualitative comparisons in Fig. 4, 5, which present SDXL results, highlight effective alignment to the reward signal, composability of our method, and robustness against over-optimization, an aspect not always captured by quantitative metrics. For instance, ReFL optimized the ImageReward signal through strategies resembling reward hacking from a human perspective. However, this

degradation was not clearly reflected in commonly used metrics such as HPS or aesthetic scores. Thus, the qualitative evaluation further underscores the value of our approach in revealing such vulnerabilities.

**Composite Reward.** As one of the evaluation tasks, we consider RePrompt-style multi-reward alignment, which imposes challenging conditions such as a few-step distillation model and object-centric prompt alignment benchmarks (Tab. 3). Our framework achieves strong qualitative and quantitative results under these settings, showing consistently high performance across an object-centric prompt alignment benchmark and multiple human-preference benchmarks. This indicates that our method effectively avoids over-optimization while achieving robust alignment. Moreover, we observe similar generalization to diffusion models unseen during training.

**Ablation Studies.** We conducted a series of ablation studies to validate the contributions of our proposed components and to analyze the effects of key hyperparameters. All experiments were performed on a single reward task (ImageReward) using the SD1.5 model. Tab. 4 summarizes the results, where each major component was added incrementally to highlight its individual effect. First, simply applying the policy model to improve prompts without training (+ policy model) degraded performance, as the model could not fully capture the task despite the use of a system prompt. Training the policy model with GRPO (+ GRPO training) led to significant improvements across all metrics. Incorporating multiple prompt refinements within a single diffusion trajectory (+ multiple improvements, 5 steps) further boosted performance. Finally, introducing visual feedback substantially increased the target reward without reducing other metrics, suggesting that it helps mitigate reward hacking (+ visual feedback).

We also investigated the impact of the number of prompt refinement steps (Fig. 6). Increasing the number of refinement steps improved not only the reward metric but also other evaluation metrics. Importantly, increasing the number of refinement steps does not increase the number of diffusion sampling steps. When trained without visual feedback, these improvements were much smaller or absent. These findings highlight that visual feedback and iterative prompt refinement are indispensable components of our equivalence MDP formulation. Together, they establish the closed-loop structure that mirrors direct RL on diffusion models, and the ablation results confirm that this formulation is not only structurally well-founded but also empirically effective.

For further analyses, including timestep-wise prompt evolution analysis and additional qualitative results, please refer to Appendix D.

## 5 CONCLUSION

In this work, we introduced PromptLoop, a plug-and-play framework for reward alignment of diffusion models via step-wise prompt refinement with latent feedback. By leveraging a multi-modal policy model trained with reinforcement learning, our method attains structural equivalence to parameter-level fine-tuning while retaining the flexibility, generality, and modularity of prompt-based alignment. Experiments demonstrate that PromptLoop achieves effective reward optimization, generalizes seamlessly to unseen diffusion backbones, composes orthogonally with existing alignment techniques, and mitigates over-optimization and reward hacking. These results position PromptLoop not only as a structurally sound but also as a practically robust complement to weight-level tuning. Overall, PromptLoop provides a simple yet effective path toward more reliable and adaptable generative models, while its plug-and-play nature facilitates integration into user-facing applications, underscoring strong potential for real-world deployment.

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

## A    RELATED WORKS

**Aligning Diffusion Models.** Following the success of RLHF for LLMs, there has been growing interest in aligning diffusion models with human preferences or arbitrary reward functions. Methods such as DDPO (Black et al., 2024), Diffusion-DPO (Wallace et al., 2024), and DanceGRPO (Xue et al., 2025) treat the diffusion sampling process as a Markov decision process (MDP), and train the diffusion model using RL algorithms. In contrast to RL-based approaches that rely on black-box rewards, other methods directly exploit the gradient of the reward or objective function. For example, ReFL (Xu et al., 2023) optimizes sampling trajectories via reward gradients, applying the reward to intermediate denoised estimates to avoid full backpropagation. ELLA (Hu et al., 2024) introduces a timestep-aware connector module that maps encoded prompt embeddings before they are fed into the diffusion model. More recently, Adjoint Matching (Domingo-Enrich et al., 2024) casts reward fine-tuning as a stochastic optimal control (SOC) problem, optimizing with reward gradients.

**Prompt-based Improvements for Diffusion Models.** In text-to-image generation, prompts serve as a powerful control signal and have been widely leveraged as a means of alignment. Prior work such as OPT2I (Mañas et al., 2024), RATTPO (Kim et al., 2025a), and TIR (Khan et al., 2025) explores LLM-based prompt refinement without fine-tuning, relying on feedback from evaluations of fully generated images to suggest improved prompts. To align LLM-based prompt refinement more closely with reward, Promptist (Hao et al., 2023), RePrompt (Wu et al., 2025), and PromptEnhancer (Wang et al., 2025a) fine-tune LLMs with reinforcement learning, treating the diffusion model simply as a black-box reward model in a feedforward manner. RL-based alignment has also been extended beyond diffusion models to autoregressive (AR) multimodal models, where methods such as Visual-CoG (Li et al., 2025) and IRGL (Huang et al., 2025) adopt CoT-style approaches that iteratively generate prompts and images through self-feedback to achieve reward alignment.

## B    DETAILED ALGORITHM

We summarize the procedure of PromptLoop in two parts. Algorithm 1 presents the training process, while Algorithm 2 details the sampling procedure.

## C    IMPLEMENTATION DETAILS

### C.1    FRAMEWORK AND TRAINING

We use Qwen2.5-VL-3B-Instruct (Bai et al., 2025) as the policy model, and Stable Diffusion 1.5 (Rombach et al., 2022) (SD1.5), XL (Podell et al., 2023) (SDXL), and XL-Turbo (Sauer et al., 2024) (SDXL-turbo) as the text-to-image diffusion backbones, with the specific model chosen according to the task setting. Generation resolution, classifier-free guidance (CFG) scale, inference steps, and sampler were set to each model's default configuration, except that we used the DDIM sampler (Song et al., 2020a) for SD1.5 and 5 sampling steps for SDXL-turbo.

For GRPO training, we build on the TRL library[1] and implement our framework on top of it. Training is performed with the GRPO algorithm using a learning rate of $5 \times 10^{-6}$, batch size 8, group size 8, and $\beta$ (the KL-regularization coefficient) set to 0.005 for single-reward training and 0 for composite-reward training, without PPO clipping (num-iterations = 1). We further apply parameter-efficient fine-tuning (LoRA) (Hu et al., 2022) using the PEFT library[2], with rank $r = 16$, scaling factor $\alpha = 64$, dropout 0.05, and updates applied to all linear projection layers in the transformer blocks. All experiments are conducted in `bf16` precision on four NVIDIA A100 80GB GPUs.

To optimize our framework, we use 2 training-prompt improvement steps and 5 sampling-prompt improvement steps. Visual feedback is resized to $256 \times 256$ from the original denoised estimates obtained during the sampling process and provided to the policy model. During sampling, we insert the built-in token `<|image_pad|>` as a placeholder to replace the visual feedback.

---

[1] https://github.com/huggingface/trl
[2] https://github.com/huggingface/peft

---

**Algorithm 1:** Training PromptLoop

---

**Input:** Policy $\pi_\theta$, diffusion denoiser $\hat{\epsilon}_\phi$, sampler $f$, prompts $p_{\text{data}}$, reward $R$, # refinement steps $N_R$, GRPO group size $G$, total steps $T$

**Output:** Reward-aligned plug-and-play policy $\pi_\theta$

**1 repeat**

**2**    Sample $q \sim p_{\text{data}}$

**3**    Sample $\mathcal{R} \sim \text{Unif}(\{ R \subseteq \{1, \ldots, T\} : |R| = N_R \})$

**4**    **for** $g \in \{1, \ldots, G\}$ **do**

**5**      $c \leftarrow q$               `// init text prompt`

**6**      $\tau^g \leftarrow []$        `// trajectory: (state, action) pairs`

**7**      Sample $x_T \sim \mathcal{N}(0, I)$

**8**      **for** $t = T, T-1, \ldots, 1$ **do**

**9**        **if** $t \in \mathcal{R}$ **then**

**10**          $s_t \leftarrow (\hat{x}_t, c, q, t)$

**11**          Sample $c \sim \pi_\theta(\cdot \mid s_t)$        `// prompt refinement`

**12**          $\tau^g$.append($s_t$);    $\tau^g$.append($c$)

**13**        **end**

         `// perform one sampler step`

**14**        Sample $z_t \sim \mathcal{N}(0, I)$

**15**        $x_{t-1} \leftarrow f(x_t, z_t, c, t)$

**16**        $\hat{x}_{t-1} \leftarrow \frac{1}{\sqrt{\bar{\alpha}_t}}\left(x_t - \sqrt{1-\bar{\alpha}_t}\, \hat{\epsilon}_\phi(x_t, t, c)\right)$

**17**      **end**

**18**      $r^g \leftarrow R(x_0, q)$            `// reward calculation`

**19**    **end**

**20**    Update $\pi_\theta$ with GRPO using $\{(\tau^g, r^g)\}_{g=1}^{G}$

**21 until** *optimization complete*

---

**Algorithm 2:** Sampling with PromptLoop

---

**Input:** Policy $\pi_\theta$, diffusion denoiser $\hat{\epsilon}_\phi$, sampler $f$, input prompt $q$, refinement steps $\mathcal{R} \subseteq \{1, \ldots, T\}$

**Output:** Reward-aligned sample $x_0$

**1** Sample $x_T \sim \mathcal{N}(0, I)$

**2** $c \leftarrow q$

**3 for** $t = T, T-1, \ldots, 1$ **do**

**4**    **if** $t \in \mathcal{R}$ **then**

**5**      $s_t \leftarrow (\hat{x}_t, c, q, t)$

**6**      Sample $c \sim \pi_\theta(\cdot \mid s_t)$        `// prompt refinement`

**7**    **end**

**8**    Sample $z_t \sim \mathcal{N}(0, I)$

**9**    $x_{t-1} \leftarrow f(x_t, z_t, c, t)$

**10**    $\hat{x}_{t-1} \leftarrow \frac{1}{\sqrt{\bar{\alpha}_t}}\left(x_t - \sqrt{1-\bar{\alpha}_t}\, \hat{\epsilon}_\phi(x_t, t, c)\right)$

**11 end**

---

## C.2 PROMPTING POLICY MODELS

The policy models used for prompt refinement are guided by the instruction shown in Fig. 7, 8. As described earlier, the policy model is conditioned on the raw user input, the previously applied improved prompt, and the current timestep. In addition, we provide auxiliary information such as the total number of timesteps and the name of the target reward function. The model is then required to output an improved prompt that is suitable for the current denoising step. For the reward specification, we only provide the name of the reward (*e.g.*, `ImageReward`, `HPSv2`), without detailed definitions. This design leaves open the possibility of using the reward identifier as a mechanism for multi-reward alignment in future work. For composite rewards, the increased complexity results in longer prompts, which can hinder the diffusion model's responsiveness. To address this, we employ a dedicated prompt design that explicitly accounts for this issue.

---

**Policy Model Prompt (Single Reward)**

**User Prompt:**
You are helping to refine a prompt for an image generation diffusion model. At each timestep, you are given the input prompt, lastly improved prompt with timestep, current timestep, total timesteps, a target reward function, and the partially generated image at the current diffusion timestep. Your task is to suggest an improved prompt that better aligns with the goal. Do not attempt to correct blurriness, as the partially generated image is expected to be unclear during diffusion.

Respond *only* with a valid JSON object in the following format without any other text:

```
{
    "improved_prompt": "<your improved prompt string>"
}
```

Input:

```
{
    "input_prompt": {input_prompt},
    "last_prompt": {applied_prompt},
    "target_reward": {target_reward},
    "current_timestep": {current_timestep},
    "total_timesteps": {total_timesteps},
}
```

---

Figure 7: Prompt provided to the policy model for refinement. The instruction specifies the available context (user input, last improved prompt, timestep information, and reward name), and the model must output an improved prompt in JSON format.

## C.3 REWARD MODELS

In the single-reward setting, we used ImageReward (Xu et al., 2023), incompressibility (Black et al., 2024), compressibility (Black et al., 2024), and aesthetic score models (Schuhmann, 2025) without any modification from their official implementations and checkpoints. For the composite reward in the RePrompt-style setting, we adopted the same components—visual reasoning, length, and structure rewards. The visual reasoning reward consists of ImageReward and a VLLM-based reward, weighted equally, where the latter is implemented with `gpt-5-mini-2025-08-07` (OpenAI, 2025). The evaluation prompt for the VLLM reward is shown in Fig. 9. This design complements ImageReward by preventing reward hacking related to weak text alignment and aesthetic biases. The length reward follows the original formulation without change, while the structure reward is adapted to match our output format (JSON). Across all reward components, the scoring ranges and configurations remain unchanged.

---

**Policy Model Prompt (Composite Reward)**

**User Prompt:**
You are helping to refine a prompt for an image generation diffusion model.

[IMPORTANT] However, you must make *minimal changes* to the original user's input and *keep the prompt as simple as possible*. I *strongly* recommend *not modifying* the input prompt if possible. [IMPORTANT]

Respond *only* with a valid JSON object in the following format without any other text:

```
{
  "improved_prompt": "<your improved prompt string>"
}
```

Input:

```
{
  "input_prompt": {input_prompt},
  "last_prompt": {applied_prompt},
  "target_reward": {target_reward},
  "current_timestep": {current_timestep},
  "total_timesteps": {total_timesteps},
}
```

---

Figure 8: Prompt provided to the policy model for refinement. The instruction specifies the available context (user input, last improved prompt, timestep information, and reward name), and the model must output an improved prompt in JSON format.

---

**VLLM Reward Model Prompt**

**User Prompt:** You are an expert evaluator of text-to-image alignment. Your primary goal is to check whether the image faithfully matches the input prompt. Pay special attention to object identity, count, attributes (such as color, size, shape), and spatial relationships.
Penalize any elements that are not requested in the prompt — unnecessary decorations, background additions, or irrelevant visual noise. Missing or incorrect objects should also lower the score.
The best images are object-centric: focused on the entities and relationships specified in the prompt, while also being visually coherent and pleasant.

Please rate this image on a scale of 0-10 (10 being perfect) and explain your reasoning. Please put your score in <score> score </score>. Prompt: {p}

---

Figure 9: Prompt template for the VLLM reward in the RePrompt-style composite setting, guiding fine-grained alignment checks and producing a structured score.

## C.4 EVALUATIONS

**Baselines.** We use the official public PyTorch implementations of DDPO[3] and ReFL[4], training them on the same dataset and reward model as PromptLoop. For ReFL on SD1.5, we perform full model fine-tuning, whereas for DDPO and ReFL on SDXL we adopt LoRA-based training. Reported performance values correspond to checkpoints where evaluation rewards match those of PromptLoop. Qwen2.5-VL-3B is incorporated without GRPO training, relying solely on prompting (including visual feedback and multi-turn refinement), while maintaining the overall framework. RePrompt is implemented by removing visual feedback and multi-turn refinement from PromptLoop; reason-

---

[3] https://github.com/kvablack/ddpo-pytorch
[4] https://github.com/zai-org/ImageReward

ing is also omitted to ensure fair comparison under equivalent conditions. For Diffusion-DPO[5] and NPNet[6], we directly used their officially released checkpoints and inference code without modification.

**Metrics.** For the single-reward setting, we evaluate models using ImageReward (Xu et al., 2023), HPSv2 (Wu et al., 2023), and an aesthetic scoring model (Schuhmann, 2025). These metrics assess prompt alignment, consistency with human preference, and robustness to over-optimization. We follow the standard evaluation protocols provided in the public implementations without any modifications.

In addition, we compute VLLM scores using a pretrained multimodal large language model, Qwen2.5-VL-3B-Instruct (Wang et al., 2024a). The evaluation is performed locally with carefully designed prompts that balance human-preference alignment and aesthetic quality. Input images are resized to $512 \times 512$ before being fed into the model. The evaluator is instructed to provide a score between 0 and 10, with 10 indicating perfect quality. Scores are subsequently normalized to the range $[0, 1]$ during post-processing. The full evaluation prompt is shown in Fig. 10.

For all these metrics, the evaluation prompts are drawn from the validation split of the Pick-a-Pic v2 dataset.

---

**VLLM Score Metric Prompt**

**User Prompt:**
You are an expert image evaluator. Your task is to judge an image based on two equally weighted aspects:

1. *Faithfulness to Prompt*: Does the image accurately reflect the user's input prompt in terms of objects, attributes, style, and composition?
2. *Aesthetic Quality*: Is the image visually appealing, well-composed, and artistically pleasant from a human perspective?

Please rate this image on a scale of 0-10 (10 being perfect) and explain your reasoning. Please put your score in <score> score </score>. Prompt: {prompt}

---

Figure 10: Evaluation prompt used for computing VLLM scores. The scoring model jointly considers prompt faithfulness and aesthetic quality, and outputs a rating from 0 to 10, which is subsequently normalized to the range [0, 1] in a post-processing step.

In the composite-reward setting, we additionally evaluate on the GenEval benchmark (Ghosh et al., 2023), which emphasizes object-centric aspects of text-to-image generation. We directly adopt the prompts and evaluation procedures provided by the GenEval benchmark without modification. When measuring ImageReward, HPSv2, we also use the prompts and the sample counts from GenEval.

# D  ADDITIONAL RESULTS

## D.1  PROMPT EVOLVEMENT ANALYSIS

Since our method controls the sampling dynamics of the diffusion model through textual prompts, the evolution trajectory over diffusion timesteps optimized via reinforcement learning remains interpretable, unlike Hu et al. (2024). To analyze this, we examine the outputs of a policy model trained on SDXL with ImageReward as a single reward signal. Tab. 5 illustrates how the optimized prompts evolve as the diffusion timesteps progress.

Not every case follows the exact same trajectory, but a consistent overall pattern emerges across examples. At early timesteps, prompts typically emphasize meta-level descriptors highlighting quality, style, and realism (*e.g.*, "photorealistic," "vivid colors"), establishing a broad atmospheric framing.

---

[5]https://github.com/SalesforceAIResearch/DiffusionDPO
[6]https://github.com/xie-lab-ml/Golden-Noise-for-Diffusion-Models

Table 5: Comparative analysis of prompt evolvement at different timesteps. Early prompts emphasize broad atmospheric qualities, intermediate prompts expand into concrete details, and later prompts either preserve these specifics or revert to prototypical descriptors.

| | Initial ($t = 981.0$) | Middle ($t = 581.0$) | Final ($t = 181.0$) |
|---|---|---|---|
| **Corgi Dog** | ...corgi wearing a hat and sunglasses, sitting on a beach chair, with a **picturesque beach and ocean in the background**. | ...corgi puppy wearing a multicolored bucket hat and sunglasses, sitting on a **plush beach chair** with its paws on the cushion, set against a background of a **vibrant sandy beach, choppy waves, and lush tropical scenery...** | ...corgi wearing a colorful straw hat and large sunglasses, sitting on a sunlit beach chair with a **tropical beach landscape, including palm trees and the ocean waves in the background**. |
| **City Night Scene** | ...lively city street at night with bright lights, towering skyscrapers, and people walking, with **vibrant colors and realistic lighting effects**, in the background there are **numerous illuminated signs and decorations**. | ...bustling city street at night with bright lights, tall buildings, and people walking, **realistic-looking photo with vibrant colors and detailed textures**. | ...lively city street at night with bright lights, tall buildings with illuminated signs, bustling crowds, and **vibrant city lights surrounding it**, **realistic photo-like scene with warm and inviting glow**. |
| **Mountain View** | ...stunning mountain landscape with snow-capped peaks, vibrant pine trees, and a clear blue sky, with **stunning lighting and vibrant colors**. | ...stunning mountain landscape with snow-capped peaks, vibrant pine trees, a clear blue sky **with fluffy clouds**, realistic photo, **warm sunset lighting**, **beautiful natural scenery**. | ...stunning mountain landscape with snow-capped peaks, vibrant pine trees, and a clear blue sky in the background, with **colorful lighting effects** and a **fluffy cloud in the sky**. |

As inference advances to intermediate timesteps, these high-level descriptors give way to more concrete and fine-grained details, such as object properties, environmental elements, or specific lighting conditions, resulting in richer and more grounded descriptions. Toward later timesteps, we observe two dominant tendencies: in some cases, prompts continue to preserve the specificity around salient elements of the scene, while in others they collapse back into prototypical atmospheric cues (*e.g.*, "warm glow," "serene atmosphere"). This overall progression—from evaluative abstraction, to concrete specificity, and finally toward either preserved details or prototypical generalities—highlights how reinforcement-learned prompt evolution balances descriptive richness with compact, high-level guidance throughout the diffusion trajectory.

Interestingly, the RL-optimized prompt evolution trajectory aligns with well-known scheduling strategies of classifier-free guidance (CFG). In diffusion models, it is established that the early steps focus on generating coarse global structures, while later steps refine finer details (Yu et al., 2023). Consistent with this, prior studies have demonstrated that applying a strong CFG too early can be harmful, leading to a variety of scheduling strategies. Two dominant families of approaches exist: those that monotonically increase CFG strength throughout the sampling process and those that increase CFG up to intermediate timesteps before decreasing it again toward the final steps (Wang et al., 2024b; Kynkäänniemi et al., 2024; Papalampidi et al., 2025). Since stronger CFG effectively enforces sharper and more detailed conditioning, our results suggest that the RL-trained policy implicitly learns both types of dynamics at the textual level, adapting prompt specificity in ways that mirror optimal CFG schedules. This emergent behavior, despite not being explicitly instructed, is intriguing.

## D.2 MORE QUALITATIVE SAMPLES

We present qualitative samples corresponding to the quantitative evaluation of single-reward alignment on SD1.5 and composite-reward alignment on SDXL-turbo reported in Tab. 2 and Tab. 3, which could not be included in the main text due to space constraints. Specifically, Fig. 11, 13 illustrates comparisons against baseline reward alignment methods, Fig. 12, 14 highlights the orthogonality of our approach to other reward alignment techniques. The results, consistent with the quantitative findings, demonstrate clear advantages in prompt alignment and human preference, while also highlighting the orthogonality and generalization capability of our approach. In addition to ImageReward as a single-reward task, we also trained models using aesthetic quality (Schuhmann, 2025), compressibility, and incompressibility rewards (Black et al., 2024), as shown in Fig. 15. These ex-

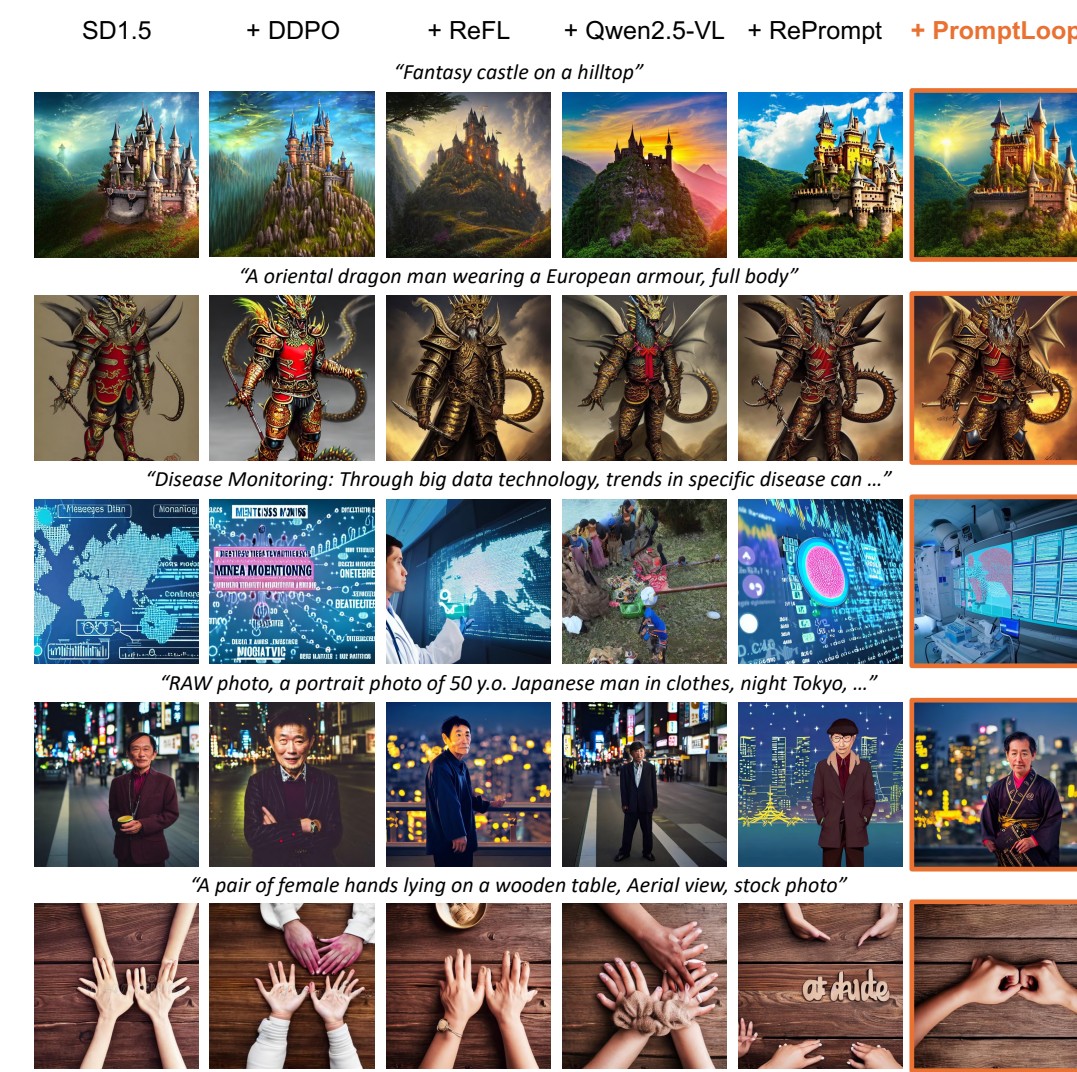

Figure 11: Qualitative comparison of baseline methods (SD1.5 & ImageReward).

periments further demonstrate that our proposed framework can be generally applied across diverse reward types.

# E  LLM USAGE

Large Language Models (LLMs) were used solely as an editorial aid to improve the clarity and readability of the manuscript. Specifically, LLMs assisted in polishing grammar, refining sentence structure, and ensuring consistency in style. They were not used in any aspect of research ideation, experimental design, data analysis, or in the generation of substantive scientific content. All ideas, results, and interpretations presented in this paper are the responsibility of the authors.

DDPO    **+ PromptLoop**    Diffusion-DPO    **+ PromptLoop**    ReFL    **+ PromptLoop**

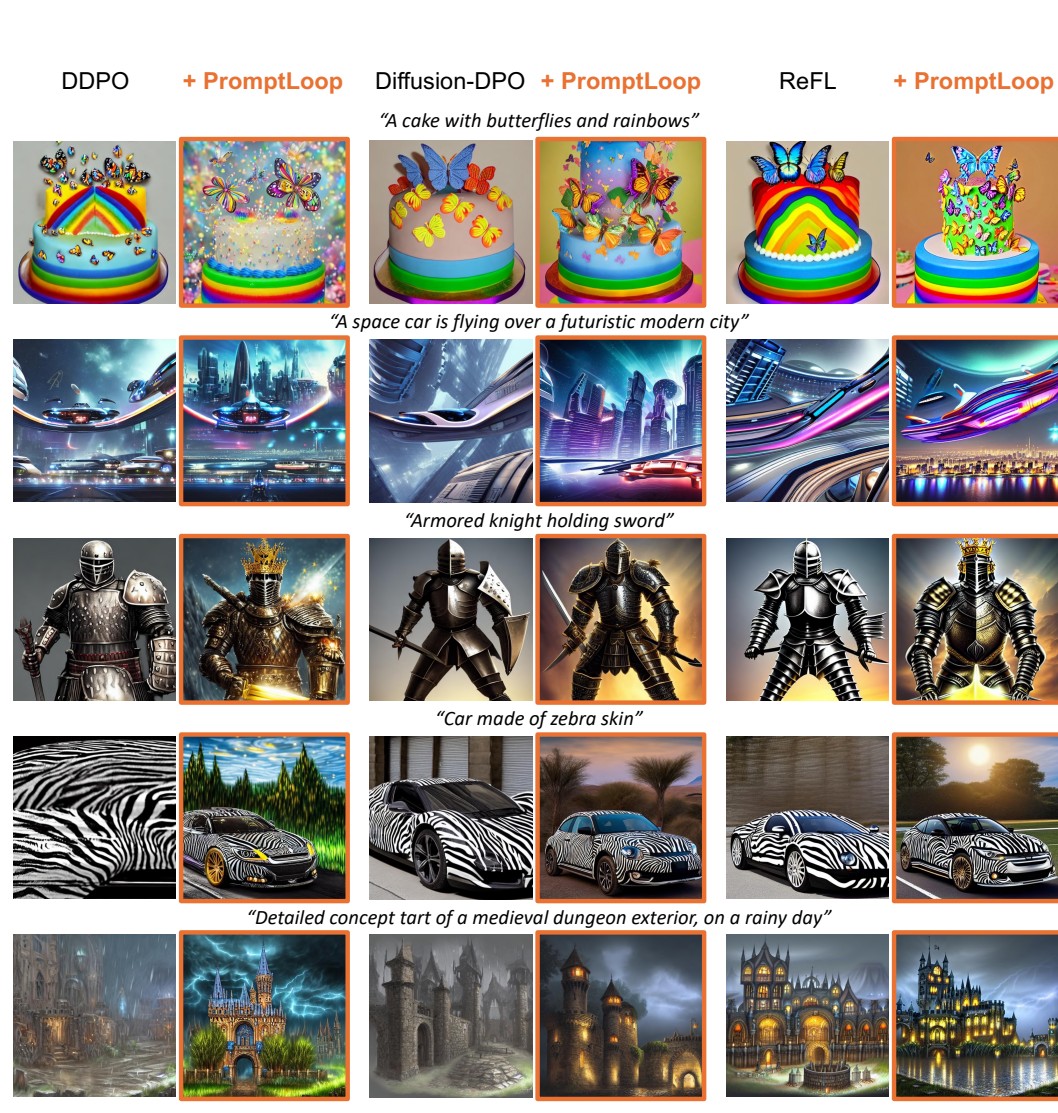

Figure 12: Qualitative results demonstrating the orthogonality of our method compared with reward-aligned baselines (SD1.5 & ImageReward).

SDXL-Turbo   + Qwen2.5-VL   + RePrompt   **+ PromptLoop**

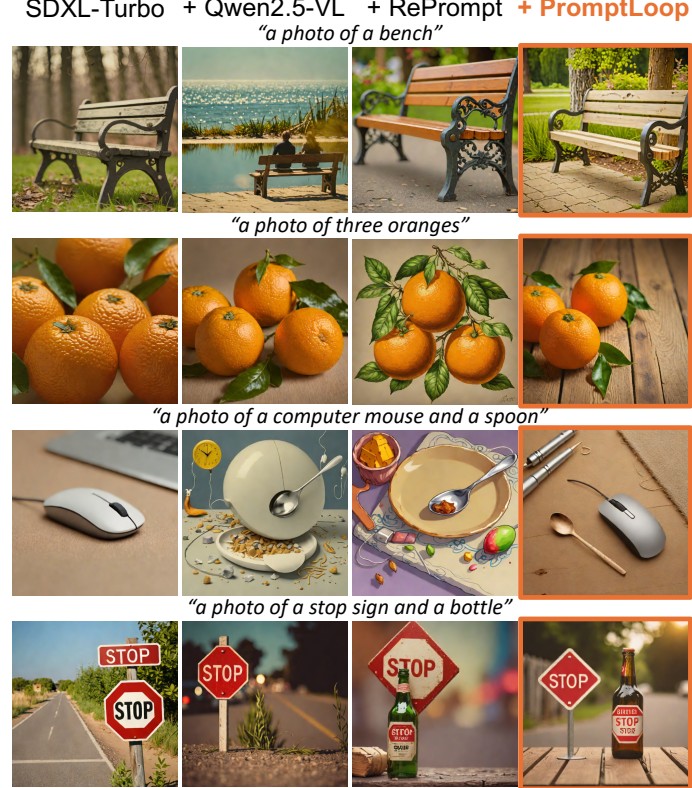

Figure 13: Qualitative comparison of composite-reward alignment, illustrating improvements over baseline methods. (SDXL-turbo & RePrompt)

SDXL   **+ PromptLoop**   SD1.5   **+ PromptLoop**

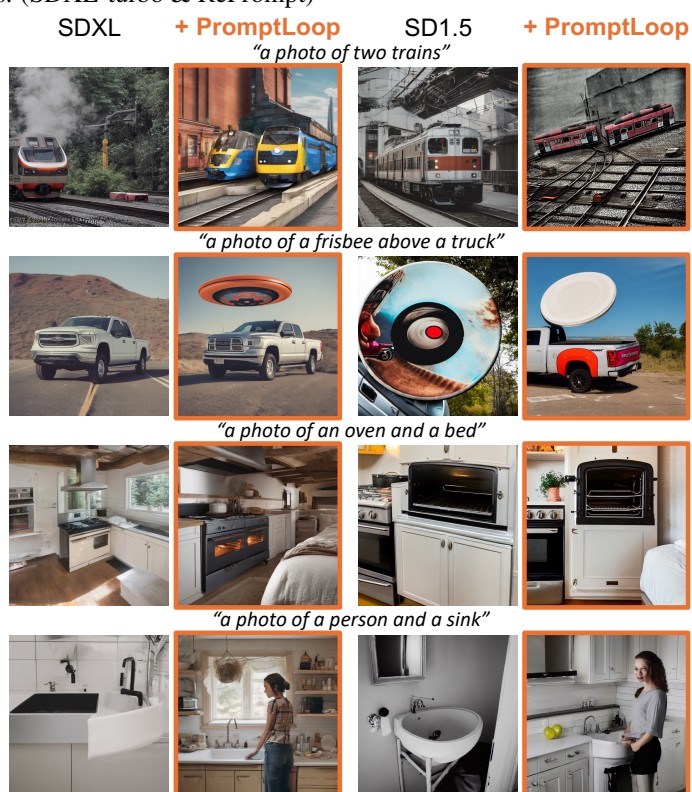

Figure 14: Qualitative results showing the orthogonality and generalizability achieved by applying our method to unseen reward-alignment baselines (SDXL-turbo & RePrompt).

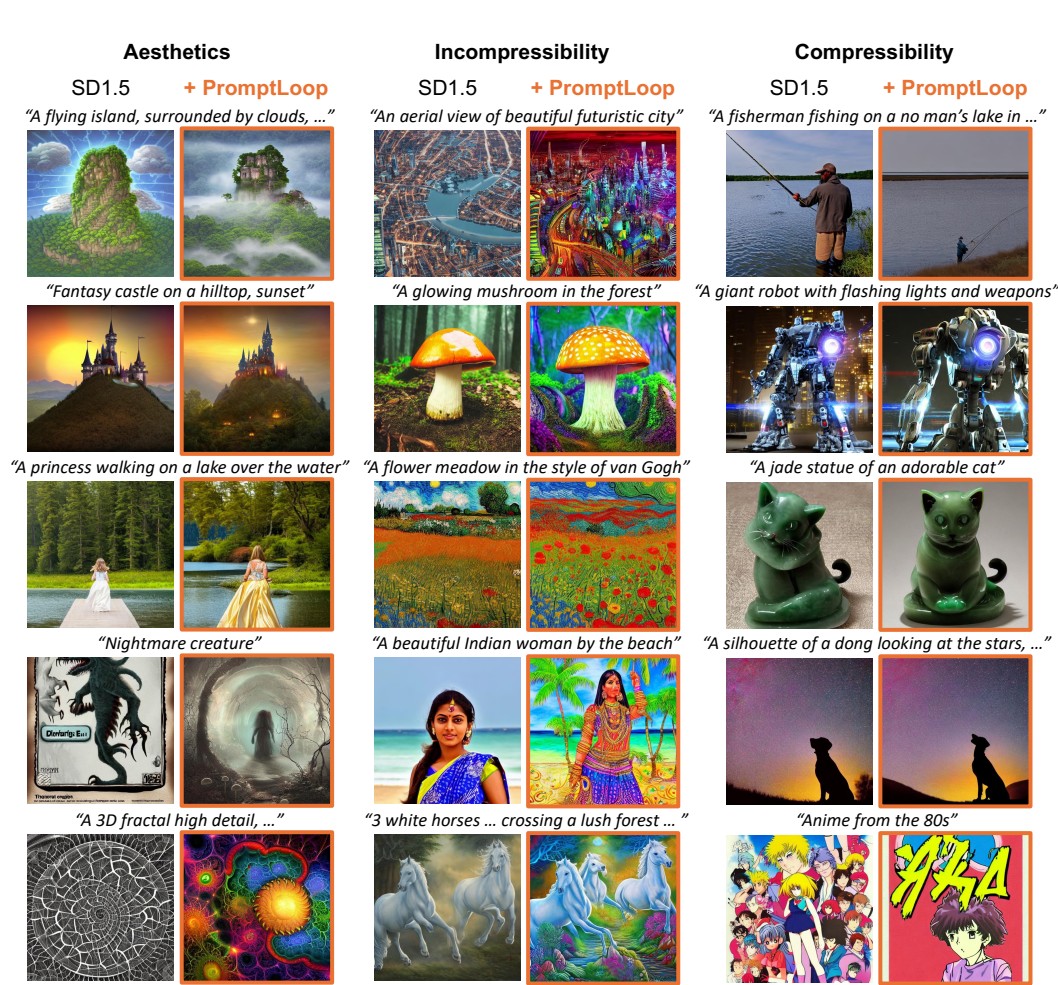

Figure 15: Qualitative results demonstrating the applicability of our framework to diverse reward signals.

