# OpenReview forum: "Plug-and-Play Prompt Refinement via Latent Feedback for Diffusion Model Alignment"
_ICLR.cc/2026/Conference — ICLR 2026 Conference Withdrawn Submission_

### Official Review · Reviewer_kBiC · 2025-10-28

**Soundness:** 2
**Presentation:** 3
**Contribution:** 3
**Rating:** 4
**Confidence:** 4

**Summary:**

The paper proposes PromptLoop, a plug-and-play RL framework that refines prompts step-wise during diffusion sampling using a policy MLLM that reads intermediate (denoised) latents. This yields a closed loop analogous to Diffusion RL while keeping the base diffusion weights frozen. The policy is trained with GRPO; to control cost, refinement is done at a sparse set of timesteps, and the authors note visual feedback is needed in training but not necessarily at inference. Experiments show modest but consistent gains over single-shot prompt engineering across standard reward/quality metrics.

**Strengths:**

1. The method layers on top of existing diffusion pipelines without touching generator weights, making adoption low-risk and compatible with multiple backbones/schedulers.

2. Iterative, timestep-aware prompt edits leverage denoising feedback to steer trajectories, offering finer control than one-shot prompt engineering.

3. Across several reward/quality metrics and backbones, the approach yields modest but reliable improvements, suggesting robustness to architectural details.

4. Because it operates purely in the text channel, it can stack with DDPO/DPO/ReFL-style methods, providing additive gains without retraining the base model.

**Weaknesses:**

1. The method is primarily empirical. The paper does not provide a formal analysis explaining how or when intermediate states $x_t$ contain sufficient semantic information to guide effective prompt refinement.

2. Predicting meaningful guidance about $x_0$ from $x_t$ is inherently difficult—the very reason why diffusion models require multiple denoising steps. It is unclear how often the VLM can extract stable, causal cues rather than reacting to transient artifacts.

4. The reported improvements over strong single-step prompt engineering baselines are modest. It is not obvious that the additional latency and system complexity are justified by the measured benefits.

5. It remains unclear whether the gains arise from (i) step-wise conditioning on intermediate latents or (ii) simply learning a better global rewrite policy with more compute. The current ablations do not fully disentangle these effects.

6. The approach relies heavily on learned preference models (e.g., ImageReward, HPS), which raises concerns about potential reward hacking or overfitting to specific evaluation models. Human evaluation is minimal.

**Questions:**

1.  When does it help most? Break down results by prompt type (composition, fine-grained attributes, rare objects). If the method mainly helps with certain failure modes (e.g., compositional binding), highlight that.

2.  How does performance change with perturbed or biased reward models? Any evidence of reward gaming (e.g., aesthetics gains but fidelity drops)?

3. Provide qualitative “prompt trajectories” across timesteps, and analyze which edits persist to the final image vs. vanish.

4. Report end-to-end generation time/VRAM for common configs and the marginal cost per additional edit step.

---

### Official Review · Reviewer_8yXn · 2025-11-01

**Soundness:** 2
**Presentation:** 3
**Contribution:** 3
**Rating:** 4
**Confidence:** 4

**Summary:**

This paper formulates T2I prompt engineering as a multi-step decision problem, where the T2I model serves as the environment and intermediate noisy images x_t provide feedback during sampling. Experiments show this approach outperforms both single-step prompt engineering and traditional RL methods.

**Strengths:**

1. Effectiveness: Figure 4 and Table 2 demonstrate clear improvements over existing methods like RePrompt
2. The method addresses an important baseline (prompt engineering) widely used in applications like nano banana, showing significant potential

**Weaknesses:**

The core concern is that the VLM receives noisy images x_t as input. While the paper uses approximation methods to predict x_0 from x_t, these approximations are known to be unreliable. Direct input of x_t to VLM is also problematic since most existing VLMs haven't seen noisy data during training, potentially causing severe out-of-distribution (OOD) issues.

The handling of this critical aspect appears highly empirical and lacks sufficient theoretical guarantees. Have the authors observed such phenomena? It would be valuable to provide concrete examples showing:

- Actual images fed to the VLM during sampling
- Corresponding VLM prompt refinements
- Visual demonstration of the refinement process

Missing Comparisons:

The paper lacks head-to-head comparison with simple prompt engineering. A more compelling evaluation would show performance vs. number of VLM refinement steps (where 1 step = traditional method), better highlighting PromptLoop's necessity over existing approaches.

**Questions:**

The idea is interesting but needs stronger empirical validation of the noisy input handling and clearer demonstration of when multi-step refinement is truly beneficial.

---

### Official Review · Reviewer_35hH · 2025-11-02

**Soundness:** 3
**Presentation:** 3
**Contribution:** 2
**Rating:** 2
**Confidence:** 4

**Summary:**

This paper introduces PromptLoop, a plug-and-play framework for aligning text-to-image diffusion models with various reward functions using reinforcement learning (RL). Instead of directly fine-tuning the diffusion model's weights, PromptLoop trains a separate multimodal large language model (MLLM) to act as a policy that iteratively refines the input prompt at different steps of the diffusion sampling process. The MLLM receives feedback from the intermediate latent states of the diffusion model, allowing for a closed-loop refinement process that is structurally analogous to direct RL-based fine-tuning of the diffusion model itself. The authors argue that this approach retains the benefits of prompt-based alignment while achieving more effective reward optimization and mitigating issues like reward hacking. The effectiveness of PromptLoop is demonstrated through extensive experiments on single and composite reward functions across different diffusion models like SD1.5 and SDXL.

**Strengths:**

- The core idea of using an MLLM to perform stepwise prompt refinement based on latent feedback is innovative. It cleverly bridges the gap between direct parameter fine-tuning (like DDPO) and feed-forward prompt optimization methods. The formulation of this process as a Markov Decision Process (MDP) where the prompt is the "action" is elegant and provides a solid theoretical grounding.
- The paper presents a thorough evaluation across multiple diffusion backbones (SD1.5, SDXL, SDXL-turbo), various baseline methods (DDPO, ReFL, RePrompt), and different reward settings (single reward like ImageReward, and composite rewards).
- The paper provides qualitative evidence suggesting that PromptLoop is more robust against over-optimization and reward hacking compared to some baselines like ReFL.

**Weaknesses:**

- The primary weakness of this method is the significant computational overhead introduced. Invoking a large MLLM multiple times within a single image generation pass is computationally expensive and increases inference latency substantially. The paper acknowledges this and proposes a "sparse refinement strategy" and an inference-time optimization where prompts are generated a priori without visual feedback. However, this optimization seems to contradict the core premise of the paper, which is the importance of latent feedback. The ablation study (Figure 6) shows that performance degrades without visual feedback.
- The success of PromptLoop heavily relies on the capabilities of the policy MLLM (Qwen2.5-VL-3B). This introduces another large, complex model into the pipeline, which may have its own biases, failure modes, and training instabilities. The framework's performance is therefore bounded by the MLLM's ability to understand the subtle changes in noisy latent states and generate meaningful prompt modifications. This dependency makes the overall system more complex and potentially less robust than self-contained fine-tuning methods.
- The authors did not compare with recent diffusion RL methods, such as FlowGRPO/DanceGRPO. They achieve significant reward improvement (e.g., 90+ score on GenEval), while also mitigating reward hacking by KL regularization. Compared to these methods, the improvement brought by PromptLoop is marginal.
- The claim of "plug-and-play" is questionable. "Plug-and-play" typically refers to training-free methods, while PromptLoop requires RL tuning of MLLMs. The generalization to unseen diffusion backbones is not notable given the marginal quality improvement.

**Questions:**

- Could you provide a more detailed analysis of the computational costs (both in terms of VRAM and time) during inference for PromptLoop compared to the baselines? Specifically, how much latency does performing 5 steps of prompt refinement add to the generation process with and without the a priori prompt generation strategy?
- The ablation study in Figure 6 shows a clear performance drop when visual feedback is removed. Given that this feedback is central to your method's novelty, how do you justify the claim that the a priori generation strategy (which omits this feedback at inference) "yields substantial generalization capability and efficiency gains while... uniquely retaining the advantages of closed-loop RL fine-tuning"? It appears to abandon the closed-loop nature of the process.
- Is there any advantage of adopting PromptLoop compared to advanced diffusion RL methods like FlowGRPO?

---

### Official Review · Reviewer_bD7z · 2025-11-05

**Soundness:** 3
**Presentation:** 3
**Contribution:** 3
**Rating:** 4
**Confidence:** 3

**Summary:**

This paper proposes a plug-and-play prompt refinement framework -- PromptLoop, which uses latent feedback (intermediate states) rather than just fixed prompts. It shows that the method generalizes across unseen diffusion models (since prompts transfer) and composes orthogonally with other alignment methods. It also provides ablations showing the benefit of step‐wise prompt updates (rather than single static prompt) and of using latent feedback.

**Strengths:**

The paper is well-written, and the idea is clear. As the prompt policy is model‐agnostic, it can in principle be applied to unseen diffusion models — a major practical benefit. Also framing prompt refinement as a sequential policy interacting with intermediate latent states is a conceptually neat bridge between prompt‐engineering and full model fine‐tuning. The experiments are suitably reasonable.

**Weaknesses:**

Several questions:

(1) What about the failure cases?

(2) The idea can also been applied to other methods, e.g., DPO. See the paper Fine-Tuning Diffusion Generative Models via Rich Preference Optimization, arXiv:2503.11720, Zhao et al., for using the VL feedback to prompts for data curation and generation.

**Questions:**

See weaknesses.

---

### Note · Authors · 2025-11-14

**Comment:**

We would like to thank the reviewers for their time and constructive feedback. After careful consideration, we have decided to withdraw our submission. We appreciate the reviewers’ efforts and thoughtful evaluations.

**Withdrawal Confirmation:**

I have read and agree with the venue's withdrawal policy on behalf of myself and my co-authors.